# TOWARDS DEBIASED SOURCE-FREE DOMAIN ADAPTATION

## ABSTRACT

Source-Free Domain Adaptation (SFDA) aims to adapt a model trained in an inaccessible source domain $S$ to a different, unlabelled target domain $T$. The crux is to remove the source bias, which leads to catastrophic failure on $T$ samples that are dissimilar to $S$. Unfortunately, without any prior knowledge about the bias, the current SFDA setting has no mechanism to circumvent it. We introduce a practical setting to address this gap—Debiased SFDA, where the model receives additional supervision from a *pre-trained*, *frozen* reference model. This setting stays in line with the essence of SFDA, which accommodates proprietary source-domain training, while also offering prior knowledge that is unaffected by source-domain training to facilitate debiasing. Under this setting, we propose 1) a simple contrastive objective that debiases the source-trained model with the guidance from the reference model. 2) a diagnostic metric that evaluates the degree to which an adapted model is biased towards $S$. Our objective can be easily plugged into different baselines for debiasing, and through extensive evaluations, we demonstrate that it engenders consistent improvements across standard benchmarks. Code is supplied under supplementary material.

## 1 INTRODUCTION

Domain adaptation (DA) aims to adapt a model trained on a labeled source domain $S$ (*e.g.*, computer-synthesized images) to generalize to a novel, unlabeled target domain $T$ (*e.g.*, real-world images), where $S$ and $T$ exhibit significant domain gap. In particular, recent studies look at *source-free domain adaptation* (SFDA), a more practical setting that provides a pre-trained source model in lieu of data, as rising concerns regarding data privacy and intellectual property render the source data unavailable (Liang et al., 2020; Chen et al., 2023). However, the source model is inevitably biased to spurious correlations that only hold in $S$ (Zhang et al., 2022). The crux of SFDA is thus to debias the source model, such that transferable, pertinent knowledge (*e.g.*, object shape) is retained as discriminative cues, and source-specific traits (*e.g.*, object orientation) irrelevant to the target domain are discarded. One common approach treats local clusters in $T$ (based on features extracted by the source model) as transferable knowledge (Yang et al., 2021a; 2022; Hwang et al., 2024). Following the intuition that the source pre-trained model has learned good class representation, this enforces prediction consistency between each $T$ sample and its $k$ nearest neighbors.

Unfortunately, debiasing a biased model without additional prior is paradoxical (Caron et al., 2020; Huang et al., 2017). Consider the aforementioned approach. The source pre-trained model may associate similar samples feature-wise based on orientation (*i.e.*, spurious) as opposed to shape (*i.e.*, transferable). The use of such spurious correlations may cause samples of different classes to have the same prediction, as they are erroneously considered among the nearest neighbors.

To show this, we design a diagnostic experiment in Figure 1, where we select test samples with confident *correct* predictions by the source model as the easy split, and those with confident *wrong* predictions as the hard split. We exhaust the experiment design space of SFDA by investigating various changes and observe that there is a consistent trade-off between model performances on the easy and hard splits. Our experiments include changing the algorithm in Figure 1(a), changing the number of epochs used in source model training in Figure 1(b), and changing the number of epochs during adaptation in Figure 1(c). We provide more details regarding the experiment in the Appendix.

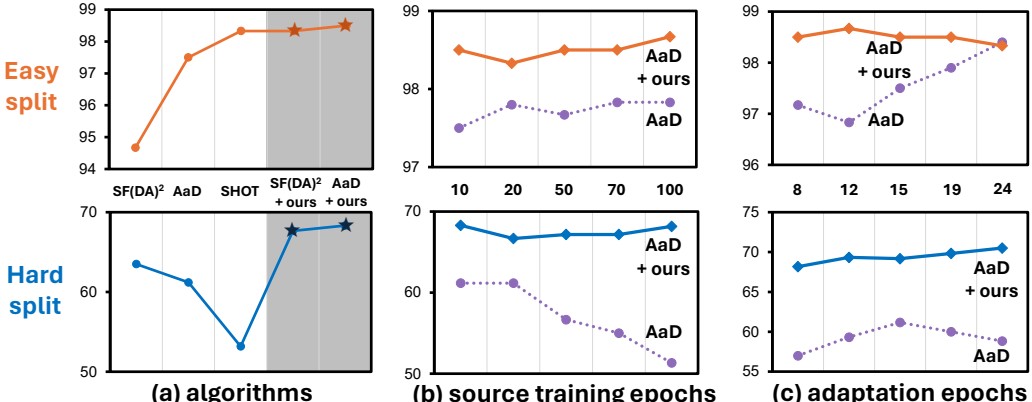

Figure 1: SFDA accuracies on an easy and hard split created out of the VisDA-C (Peng et al., 2017) test set. We change 3 key design aspects of SFDA in (a), (b) and (c). Current SFDA methods display a clear performance trade-off between the easy and hard splits (except for early adaptation epochs when the model has not converged). In contrast, adding our debiasing objective significantly improves performance on both splits.

This behavior bears strong similarity to the precision-recall trade-off in machine learning: a model either predicts too liberally and over-identifies correct samples, or too conservatively and risks missing positive samples. This trade-off can be attributed to the lack of knowledge on spurious features (Yue et al., 2021), as reliance on them results in increased false positives and/or false negatives. Similarly, without the same type of prior during target adaptation, a model either excels with source-like samples in $T$ by exploiting source-specific (*i.e.*, spurious) knowledge, or prioritize performance on difficult samples by over-adjusting to $T$ and risk losing transferable knowledge. This analysis outlines a fundamental flaw of conventional SFDA: any adapted model is simply a point on the trade-off curve and performs unreliably on classifying samples of arbitrary difficulty. It highlights the dangers of deploying such models that abuse spurious correlations into the wild, especially when it involves use cases that risk perpetuating harmful prejudices (Ji et al., 2020; Dimanov et al., 2020).

To address this deficiency, we propose a *practical* setting—*Debiased* SFDA—where a *frozen* reference model $f_r$, pre-trained on a diverse dataset (*e.g.*, ImageNet classification), provides additional supervision during the adaptation of the source model $f_s$. We highlight our rationales:

- *Practical*. Open-source pre-trained models are widely available. In real-world SFDA scenarios, such models can be leveraged to improve the performance of a proprietary $f_s$.

- *Debiased*. $f_r$ is pre-trained on large, diverse datasets, granting it strong generalization abilities. They offer useful debiasing cues by identifying predictions conflicting $f_s$ (Yue et al., 2023). For instance, an image of a car and a truck photographed in similar orientations may be deemed similar by the biased $f_s$ but dissimilar by $f_r$, suggesting that orientation is a spurious attribute.

- *Frozen*. $f_r$ *does not* receive any (spurious) supervision from $f_s$ in order to preserve the general knowledge. This is a key distinction from the ensemble approach implemented in Zhang et al. (2022), where $f_r$ is updated by the pseudo-labels generated by $f_s$ (Figure 2(c)).

Under the proposed setting, we introduce a simple contrastive objective that leverages $f_r$ to debias $f_s$. We term our approach Debiased Contrastive learning (**DeCo**), which first clusters samples in $T$ using the general knowledge of $f_r$, and then adapts the classifier of $f_s$ to align with the general knowledge, such that samples with similar predictions should come from the same cluster.

For the first step, we employ an unsupervised clustering algorithm used in generalized category discovery (Vaze et al., 2022; Wen et al., 2023) and leverage the following concepts: (a) the number of $T$ classes, which is known in SFDA, and (b) classes in $T$ can be considered novel to $f_r$, as it is pre-trained on a different dataset. Next, we take inspiration from the InfoNCE loss (van den Oord et al., 2019) and implement it such that for each training sample, positive keys in a mini-batch are defined as those from the same cluster, and negative keys as those from a different cluster. We also further propose a diagnostic metric, Hardness Accuracy (**HAcc**), to quantitatively measure an adapted model's accuracy in relation to initial source bias (*i.e.*, after debiasing). Our contributions:

- We highlight the deficiency of the current SFDA setting and propose a more practical debiased setting in Section 3.2 that is easily expandable.

- We introduce DeCo in Section 4, a simple, plug-and-play method that brings consistent improvements. As Figure 1 shows, DeCo stabilizes model performance on the easy and hard splits, effectively alleviating the trade-off issue. Our metric further corroborates this claim.

- We validate the proposed method across various benchmarks and introduce a novel **HAcc** metric to assess source bias. Our results show an improvement on VisDA by 2.4% and 4.8% in accuracy and average HAcc, respectively. We emphasize that our approach is not designed to pursue the highest accuracy on benchmarks, but to address the core issue of SFDA – bias.

## 2 RELATED WORK

**Domain adaptation.** DA aims to improve performance on unlabeled target domains by leveraging knowledge learned from a labeled source domain; the main challenge at hand is the reduction of domain shift. Conventional DA methods assume that there is access to both the source and target domain data during the adaptation process. Unsupervised domain adaptation (UDA) methods mainly fall under two categories: domain alignment methods, which seek align the source and target domain distributions (Cui et al., 2020; Liu et al., 2021a), and self-training methods, which utilize adversarial learning to learn domain-invariant feature representations (Long et al., 2018; Tzeng et al., 2017; Shu et al., 2018).

**Source-free domain adaptation.** In SFDA, the pre-trained source model is adapted to an unlabeled target dataset. There are two main families of approaches to SFDA: model-based methods, which consists of fine-tuning modular models for domain adaptation, and data-based methods, which follows the intuition of mimicking or exploring pertinent source domain data properties (Yu et al., 2023). We focus on the latter and specifically study neighborhood clustering for SFDA. SHOT (Liang et al., 2020) freezes the source classifier and clusters target features using information maximization and self-supervised pseudo-labeling. G-SFDA (Yang et al., 2021b), NRC (Yang et al., 2021a), and AaD (Yang et al., 2022) utilize local neighborhood features in the feature space to enforce consistency in predictions. Following this line of work, SF(DA)$^2$ (Hwang et al., 2024) leverages data augmentations by building augmentation graphs and identifying class partitions with spectral neighborhood clustering. Zhang et al. (2022) adopts an ensembling approach to improve pseudo-label reliability by integrating a pre-trained network during target adaptation.

**Spurious correlation in domain adaptation.** A model that is trained on a source domain is likely to learn spurious correlation in the data and subsequently perform poorly on target domains. To reduce spurious correlation, ATDOC (Liang et al., 2021) and CST (Liu et al., 2021a) propose to generate target-oriented pseudo-labels using an auxiliary classifier. Notably, these methods initialize the aforementioned classifier with source-trained classifiers, the latter of which brings both the knowledge and spurious correlation of the source domain to the former. ICON (Yue et al., 2023) eliminates this issue by employing a randomly-initialized cluster head that solely learns from the target distribution.

**Contrastive SFDA.** Contrastive learning has become an increasingly popular approach in SFDA (VS et al., 2023). VS et al. (2023) uses an instance relation graph network that exploits objects relations to guide contrastive loss. SiLAN (Wang et al., 2024) contrasts augmented views in the latent space. UC-SFDA (Chen et al., 2023) uses evidence theory to find high-confidence pseudo-labels for contrasting. Our method extends the work of ICON (Yue et al., 2023), which is designed for debiasing in the UDA setting and requires access to source data during training. We overcome this requirement by contrasting feature similarity of in-domain samples between the source-trained model and a large-scale pre-trained model.

**Bias metrics.** Studies have established numerous methods to quantify the unfairness of machine learning models (Brodersen et al., 2010; Besse et al., 2020; Mehrabi et al., 2021). We propose HAcc as we are concerned with measuring performance changes based on the reduction of *source-specific* biases as opposed to the correction of statistical biases in general. This provides a more tapered view for assessment.

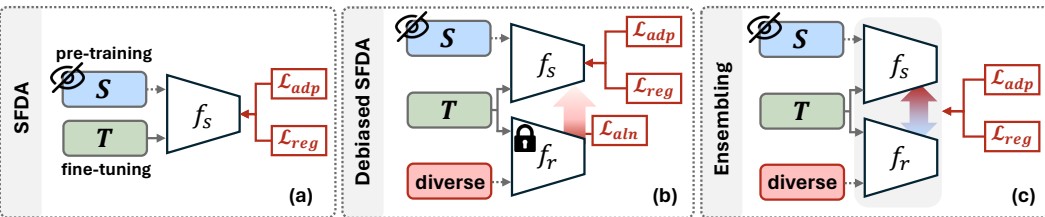

Figure 2: The paradigm of (a) SFDA with proprietary source domain training; (b) debiased SFDA where the source model $f_s$ is guided by a pre-trained, frozen reference model $f_r$; (c) ensemble, where $f_r$ also receives guidance from $f_s$, and their ensemble is used for optimization.

## 3 FORMULATION

### 3.1 SOURCE-FREE DOMAIN ADAPTATION (SFDA)

The goal is to adapt a model trained in a source domain $S$ (*i.e.*, source model) to a target domain $T$, using only an unlabelled dataset $\mathcal{D} = \{\mathbf{x}_i\}_{i=1}^N$ from $T$, where $\mathbf{x}_i$ is the $i$-th sample (*e.g.*, an image). Its paradigm is illustrated in Figure 2(a). We focus on the closed-set setting in this paper, under which the target domain shares the same $C$ classes as the source domain ($C$ is known). The source model $f_s$ consists of a feature extractor that maps a sample $\mathbf{x}$ to the feature space, and a classifier that maps the feature to its softmax-normalized probability of belonging to each class. The objective of existing methods comprise of two parts: an adaptation loss $\mathcal{L}_{adp}$ to adjust $f_s$ to the target distribution, and a regularization loss $\mathcal{L}_{reg}$ to prevent model degeneration. We detail the two losses next.

**Existing approach**. We take a representative approach AaD (Yang et al., 2022) as an example. It fine-tunes $f_s$ by minimizing the following loss for each $\mathbf{x} \in \mathcal{D}$:

$$\mathcal{L}_{aad}(\mathbf{x}) = - \overbrace{\sum_{\mathbf{x}' \in \mathcal{K}(\mathbf{x})} f_s(\mathbf{x})^T f_s(\mathbf{x}')}^{\mathcal{L}_{adp}} + \lambda \overbrace{\sum_{\mathbf{x}' \in \mathcal{Q}(\mathbf{x})} f_s(\mathbf{x})^T f_s(\mathbf{x}')}^{\mathcal{L}_{reg}}, \tag{1}$$

where $\mathcal{K}(\mathbf{x})$ is the set of $k$ nearest neighbors measured by cosine similarity in the feature space, $\mathcal{Q}(\mathbf{x})$ is given by $\mathcal{D} \setminus \mathbf{x}$ and $\lambda$ is the regularization weight. In particular, AaD assumes that $f_s$, with source training, has learned transferable feature, *i.e.*, $\mathbf{x}$ and its nearest neighbors $\mathcal{K}(\mathbf{x})$ should come from the same class, hence its adaption loss increases the dot product similarity between their predictions. However, this objective could lead to a collapsed solution, *e.g.*, $\mathcal{L}_{adp}$ is also minimized by mapping all samples in $\mathcal{D}$ to the same feature. Hence, to prevent this, $\mathcal{L}_{reg}$ encourages different samples ($\mathbf{x}$ and any sample in $\mathcal{Q}(\mathbf{x})$) to have different predictions. The two loss components can take different forms in other SFDA methods: *e.g.*, $\mathcal{L}_{adp}$ can be implemented as a cross-entropy loss calculated from the prediction $f_s(\mathbf{x})$ and pseudo-label $\arg\max f_s(\mathbf{x})$ (Chu et al., 2022; Li et al., 2022), and $\mathcal{L}_{reg}$ can be implemented by an entropy term to encourage the diversity of predictions over $\mathcal{D}$ (Liang et al., 2020).

Unfortunately, we have shown in Figure 1 that the adapted $f_s$ is still prone to spurious correlations in $S$, which cannot be resolved within the current design space of SFDA. This motivates us to propose a more practical setting in Section 3.2.

### 3.2 DEBIASED SFDA

As shown in Figure 2, unlike (a) SFDA, where the adaptation uses only a single source model $f_s$ trained on hidden source domain $S$, (b) debiased SFDA additionally leverages a reference model $f_r$, which satisfies the following points: 1) It is an open-source model, hence is generally available. 2) Its pre-training dataset should be diverse to learn general knowledge, while also related to the SFDA task to allow knowledge transfer. 3) During adaptation, $f_r$ should be frozen to prevent it from absorbing spurious correlation in $S$ (from the source model) and $T$. In this work, we drop the pre-trained classifier in $f_r$, and considers it as a mapping from the sample space to a feature space.

In particular, debiased SFDA is fundamentally different from the previous ensemble approach (Zhang et al., 2022) shown in Figure 2(c). In debiased SFDA, the adaptation of $f_s$ receives guidance from

Figure 3: The pipeline of our DeCo. We first train a cluster head $g$ using the features from the reference mdoel $f_r$ (left). Then for each sample $\mathbf{x}$, we make its prediction similar (by dot-product) to that of $\mathbf{x}_p$ from the same cluster, and dissimilar to that of any $\mathbf{x}_n$ from a different cluster.

$f_r$ to remove bias that is unaligned with its general knowledge (through the alignment loss $\mathcal{L}_{aln}$), but *not* the other way round. In contrast, the ensemble approach initializes a classifier for $f_r$ using the pseudo-labels provided by $f_s$, and both of their outputs are used to compute the adaptation and regularization loss. By interacting between $f_s$ and $f_r$, the knowledge of $f_r$ provides additional classification cues, yet it is impossible to remove the source bias in $f_s$. While ensemble can be a useful tool in performance-driven scenarios, we specifically focus our study on debiasing in SFDA.

## 4 METHOD

Based on the debiased SFDA setting, we propose Debiased Contrastive (DeCo) learning, which has two steps: 1) cluster the samples in $\mathcal{D}$ based on the features extracted by the reference model $f_r$; 2) then use a loss inspired by InfoNCE (van den Oord et al., 2019) to align the source model $f_s$ with the clusters. Our overall pipeline is depicted in Figure 3. We detail each step below.

### 4.1 CLUSTERING

**Cluster head**. Our cluster head $g$ consists of two parts, an MLP that maps the feature space of $f_r$ to a projection feature space, and a linear head that classifies a projected feature into a prediction, *i.e.*, $\mathbf{p}_i = g(f_r(\mathbf{x}_i))$ where $\mathbf{p}_i \in \mathbb{R}^C$ is the softmax-normalized probabilities of belonging to each of the $C$ clusters, and $\mathbf{x}_i \in \mathcal{D}$ is a sample. The cluster head $g$ is randomly initialized during training.

**Training of the cluster head**. Inspired by SimGCD (Wen et al., 2023), which is originally proposed for novel class discovery, we adapt the unsupervised objective of the algorithm as our cluster training objective. Specifically, we implement an unsupervised representation loss and self-distillation loss. The loss for training $g$ per mini-batch $\mathcal{B}$ in $\mathcal{D}$ is given by:

$$\mathcal{L}_g(\mathcal{B}) = \frac{1}{|B|} \sum_{i \in \mathcal{B}} \mathcal{L}_{con}(\mathbf{p}_i, \mathbf{p}'_i) + \overbrace{\frac{1}{|B|} \sum_{i \in \mathcal{B}} \mathcal{L}_{ce}(\mathbf{q}'_i, \mathbf{p}_i) - \alpha H(\bar{\mathbf{p}})}^{\mathcal{L}_{self-distillation}}, \quad (2)$$

where $\mathcal{L}_{con}$ is the self-supervised contrastive loss (Chen et al., 2020), $\mathbf{p}'_i = g(f_r(\mathbf{x}'_i))$ is the cluster prediction of the $i$-th sample in $\mathcal{B}$ under strong augmentations $\mathbf{x}'_i$, $\mathcal{L}_{ce}$ is the cross-entropy loss, $\mathbf{q}'_i$ is the sharpened predictions of $\mathbf{x}'_i$, $\alpha$ is a weight coefficient fixed to 2.0, and $H(\bar{\mathbf{p}})$ denotes the entropy of the mean batch prediction. The first term disentangles the projection feature space from augmentation-related attributes (Wang et al., 2021). Subsequently, the following terms enforce self-alignment, ensuring that predicted clusters are determined with high confidence and are strongly diverse. Overall, this unsupervised loss encourages $g$ to use class attributes invariant to strong augmentations for clustering. Further details on the clustering objective are in the Appendix.

### 4.2 TRAINING AND INFERENCE

**Alignment loss $\mathcal{L}_{aln}$.** We aim to align the source model $f_s$ with the clusters identified from the features of $f_g$. Specifically, we use a contrastive loss such that the predictions of $f_s$ are similar for a pair of samples from the same cluster, and dissimilar for a pair from different ones. The intuition is to calibrate the classification boundary of $f_s$ to lie on the low-density regions in the feature space

of $f_r$, hence $f_s$ is forced to classify based on the class attributes identified by $f_r$. Inspired by the InfoNCE loss (van den Oord et al., 2019), we use cluster identity as opposed to augmentations to compare sample similarity:

$$\mathcal{L}_{aln}(\mathbf{x}) = -\log \sum_{\mathbf{x}_p \in \mathcal{K}(\mathbf{x},\beta_1)} \frac{\exp(f_s(\mathbf{x})^T f_s(\mathbf{x}_p))}{\sum_{\mathbf{x}_n \in \mathcal{Q}(\mathbf{x},\beta_2)} \exp(f_s(\mathbf{x})^T f_s(\mathbf{x}_n))}, \tag{3}$$

where $\mathcal{K}(\mathbf{x}, \beta_1) = \{\mathbf{x}_p \in \mathcal{D} \mid g(\mathbf{x}_p)^T g(\mathbf{x}) > \beta_1\}$ selects a subset of $\mathcal{D}$ that is likely (controlled by $\beta_1$) to come from the same cluster as $\mathbf{x}$, and $\mathcal{Q}(\mathbf{x}, \beta_2) = \{\mathbf{x}_n \in \mathcal{D} \mid g(\mathbf{x}_n)^T g(\mathbf{x}) \leq \beta_2\}$ selects those which are opposite (from different clusters). We show in Section 5 the simple process of selecting $\beta_1, \beta_2$. Besides measuring the dot product similarity, *i.e.*, $g(\mathbf{x}_p)^T g(\mathbf{x})$ (referred to as "prob" contrastive mode), we define two other contrastive modes "sim" and "stats". The three modes are described and ablated in Section 5.3.

**Training and inference.** Our proposed $\mathcal{L}_{aln}$ can be plugged into existing SFDA baselines to conduct debiasing. Returning to the aforementioned representative approach, the overall training objective is given by:

$$\min_g \mathcal{L}_g(\mathcal{D}), \qquad \min_{f_s} \sum_{\mathbf{x} \in \mathcal{D}} \mathcal{L}_{AaD}(\mathbf{x}) - \mathcal{L}_{aln}(\mathbf{x}), \tag{4}$$

where $\mathcal{L}_g$ is given in Eq. (2) and $\mathcal{L}_{AaD}$ is given in Eq. (1). We slightly abuse the notation to put the function symbols as the target of optimization. During inference, only $f_s$ is utilized.

## 5 EXPERIMENTS

### 5.1 SETTINGS

**Datasets.** We evaluate our method against three benchmark datasets for image classification: Office-Home (Venkateswara et al., 2017), VisDA-C 2017 (Peng et al., 2017), and DomainNet-126 (Peng et al., 2019). **Office-Home** consists of 4 domains (Art, Clipart, Product, Real) with 65 classes per domain and 15,500 images. **VisDA** evaluates synthesis-to-real object recognition across 12 classes, containing 152,000 synthetic source images and 55,000 real target images. **DomainNet** is a large-scale benchmark with 6 domains and 345 classes. Following Chen et al. (2022), we select 4 domains (Clipart, Real, Sketch, Painting) containing 126 classes.

**Implementation details.** We conduct experiments under similar settings as described in previous work for fair comparison (Liang et al., 2020; Yang et al., 2022; Hwang et al., 2024). We follow Hwang et al. (2024) as closely as possible when evaluating against DomainNet. The architecture consists of a feature backbone and the classification head, which is a fully-connected layer with weight normalization. ResNet-50 is adopted as the backbone for Office-Home and DomainNet, and ResNet-101 is used for VisDA. We adopt SGD with momentum 0.9 and train 40 epochs for Office-Home, and 15 epochs for VisDA and DomainNet. Batch size is set to 64 for Office-Home and VisDA, and 128 for DomainNet. We adhere to the standard practice of training the last two layers of the model at a learning rate 10 times of the backbone – the learning rate is set to 1e-3 for Office-Home, 1e-4 for VisDA, and 5e-5 for DomainNet. We set the learning rate to 0.1 for $g$ across all benchmarks. For fairness, all reference models used in our experiments are pre-trained on the ImageNet-1K dataset to match the feature backbone in the source model.

**H-scores and HAcc.** As there are currently no standardized measures to quantify model bias vis-à-vis source bias for SFDA, we propose Hardness Accuracy (HAcc), a novel metric that finds the mean-class accuracy across $K$ unequal (in size and class distribution) partitions of $T$ test samples. Firstly, we begin by defining the hardness score (*i.e.*, H-score) of a sample $\mathbf{x}_i$ as the degree of uncertainty to which $f_s$ predicts $\mathbf{x}_i$ as the ground-truth label $y_i$; it is represented as $h(\mathbf{x}_i) = 1 - \phi(f_s(\mathbf{x}_i))[idx_{y_i}]$, where $\phi$ refers to the softmax function and $idx_{y_i}$ the $y_i$-th index. In other words, the higher the H-score, the more uncertain $f_s$ evaluates $\mathbf{x}_i$ as the correct label. To generate unequal splits, sample ratio per partition is determined based on the H-score using the Power Law (Clauset et al., 2009). Specifically, each subsequent split contains fewer samples of higher H-score than the previous, *i.e.*,

Table 1: Results (%) on the Office-Home dataset. [†] denotes reproduced results.

| Method | A→C | A→P | A→R | C→A | C→P | C→R | P→A | P→C | P→R | R→A | R→C | R→P | Avg |
|---|---|---|---|---|---|---|---|---|---|---|---|---|---|
| Source only | 43.5 | 67.1 | 74.2 | 51.5 | 62.2 | 63.3 | 51.4 | 40.7 | 73.2 | 64.6 | 45.8 | 77.6 | 59.6 |
| SHOT | 57.1 | 78.1 | 81.5 | 68.0 | 78.2 | 78.1 | 67.4 | 54.9 | 82.2 | 73.3 | 58.8 | 84.3 | 71.8 |
| G-SFDA | 57.9 | 78.6 | 81.0 | 66.7 | 77.2 | 77.2 | 65.6 | 56.0 | 82.2 | 72.0 | 57.8 | 83.4 | 71.3 |
| NRC | 57.7 | 80.3 | 82.0 | 68.1 | 79.8 | 78.6 | **78.6** | **65.3** | 56.4 | **83.0** | 71.0 | 58.6 | 72.2 |
| AaD | 59.3 | 79.3 | 82.1 | 68.9 | 79.8 | 79.5 | 67.2 | 57.4 | 83.1 | 72.1 | 58.5 | 85.4 | 72.7 |
| + DeCo-RN50 | 59.1 | 80.1 | 82.7 | 69.8 | 81.7 | 80.3 | 68.7 | 58.4 | 82.4 | 72.2 | 59.6 | 85.9 | 73.4 |
| + DeCo-SwinB | **64.7** | **82.2** | **85.4** | **72.0** | **84.6** | **84.7** | 71.2 | 63.9 | **85.3** | 74.0 | 65.7 | **87.8** | 76.8 |
| SF(DA)²[†] | 57.0 | 77.5 | 80.9 | 66.0 | 78.1 | 77.5 | 63.8 | 53.6 | 80.9 | 70.2 | 57.9 | 84.1 | 70.6 |
| + DeCo-RN50 | 57.0 | 77.1 | 80.5 | 67.4 | 78.5 | 77.1 | 66.8 | 54.2 | 80.1 | 71.5 | 59.2 | 83.2 | 71.0 |
| + DeCo-SwinB | 58.8 | 77.9 | 81.2 | 67.8 | 78.7 | 79.1 | 66.8 | 55.7 | 81.2 | 72.2 | 59.5 | 84.6 | 72.0 |

Table 2: Results (%) on the VisDA-C dataset.

| Method | plane | bcycl | bus | car | horse | knife | mcycl | person | plant | sktbrd | train | truck | per-class |
|---|---|---|---|---|---|---|---|---|---|---|---|---|---|
| Source only | 51.5 | 15.3 | 43.4 | 75.4 | 71.2 | 6.8 | 85.5 | 18.8 | 49.4 | 46.4 | 82.1 | 5.4 | 45.9 |
| SHOT | 94.3 | 88.5 | 80.1 | 57.3 | 93.1 | 94.9 | 80.7 | 80.3 | 91.5 | 89.1 | 86.3 | 58.2 | 82.9 |
| G-SFDA | 96.1 | 88.3 | 85.5 | 74.1 | 97.1 | 95.4 | 89.5 | 79.4 | 95.4 | 92.9 | 89.1 | 42.6 | 85.4 |
| NRC | 96.8 | 91.3 | 82.4 | 62.4 | 96.2 | 95.9 | 86.1 | 80.6 | 94.8 | 94.1 | 90.4 | 59.7 | 85.9 |
| AaD | 96.8 | 89.3 | 83.8 | 82.8 | 96.5 | 95.2 | 90.0 | 81.0 | 95.7 | 92.9 | 88.9 | 54.6 | 87.3 |
| + DeCo-RN101 | 97.4 | 88.6 | 83.1 | 79.3 | 96.8 | **97.9** | 92.2 | 83.3 | 96.1 | 91.9 | 91.5 | 54.7 | 87.7 |
| + DeCo-SwinB | **98.0** | 90.3 | **87.3** | 79.0 | **98.1** | 97.8 | **94.1** | 86.6 | **97.9** | 93.9 | **94.3** | 58.8 | 89.7 |
| SF(DA)² | 96.8 | 89.3 | 82.9 | 81.4 | 96.8 | 95.7 | 90.4 | 81.3 | 95.5 | 93.7 | 88.5 | 64.7 | 88.1 |
| + DeCo-RN101 | 97.5 | 91.0 | 85.1 | 83.2 | 97.4 | 97.5 | 92.0 | 84.1 | 96.9 | 93.5 | 91.7 | 54.8 | 88.7 |
| + DeCo-SwinB | 97.9 | **92.4** | 86.4 | **83.8** | 97.7 | 97.7 | 92.8 | 84.2 | 97.1 | **95.4** | 93.4 | **68.4** | **90.6** |

Table 3: Results (%) on the DomainNet-126 dataset. [†] represents reproduced results.

| Method | S→P | C→S | P→C | P→R | R→S | R→C | R→P | Avg |
|---|---|---|---|---|---|---|---|---|
| Source only | 50.1 | 46.9 | 53.0 | 75.0 | 46.3 | 55.5 | 62.7 | 55.6 |
| SHOT | 66.1 | 60.1 | 66.9 | 80.8 | 59.9 | 67.7 | 68.4 | 67.1 |
| NRC | 55.7 | 58.6 | 64.5 | 82.3 | 58.4 | 65.2 | 68.2 | 66.1 |
| AaD | 65.4 | 54.2 | 59.8 | 81.8 | 54.6 | 60.3 | 68.5 | 63.5 |
| + DeCo-RN50 | 66.2 | 56.6 | 61.2 | 79.2 | 56.2 | 61.7 | 66.9 | 64.0 |
| + DeCo-SwinB | **68.7** | 60.3 | 65.4 | 81.4 | 59.5 | 66.9 | 69.2 | 67.3 |
| SF(DA)²[†] | 66.6 | 54.4 | 61.2 | 79.6 | 55.5 | 62.7 | 63.9 | 63.4 |
| + DeCo-RN50 | 68.0 | 59.8 | 65.8 | 81.0 | 59.0 | 66.9 | 69.8 | 67.2 |
| + DeCo-SwinB | **68.7** | **60.5** | **66.1** | **82.3** | **60.4** | **67.8** | **71.1** | **68.1** |

earlier splits are easier *relative to* $f_s$ as they are more source-aligned. Then, HAcc is determined by averaging the mean-class accuracy across all splits. We use the VisDA-C test set to create $K = 20$ partitions as shown in Figure 4. The partition strategy is designed such that it simulates benchmarks in bias-sensitive applications (Koh et al., 2021; Zhang et al., 2024), and it is worth noting that created splits have long-tailed distributions. Additional details is provided in the Appendix.

## 5.2 Main Results

We assess the efficacy of DeCo by appending it to two state-of-the-art baselines, AaD (Yang et al., 2022) and SF(DA)² (Hwang et al., 2024), and compare it with other methods also based on neighborhood clustering. Our results on classic SFDA benchmarks are reported in Tables 1, 2, and 3. We **bold** the best performance and underline the second best.

**Standard accuracy.** We observe that DeCo provides marginal gains in average benchmark performance when $f_r$ is identical to the backbone in $f_s$. For AaD, improvement ranges from 0.4 to 0.7%, and for SF(DA)², 0.4 to 3.8% . Notably, consistent improvements across benchmark tasks are not

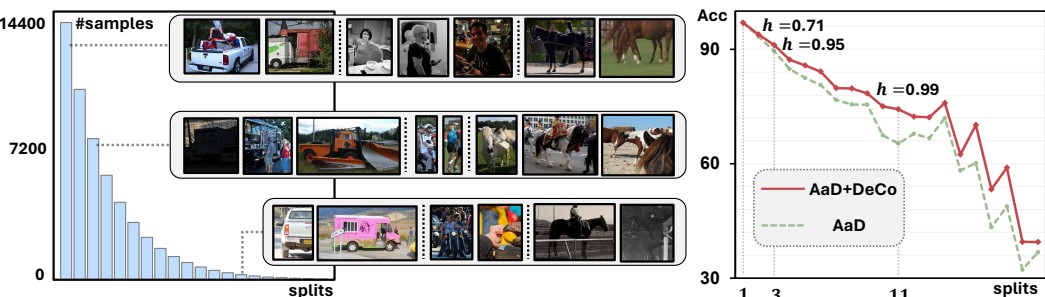

Figure 4: **Left**: The VisDA validation set is partitioned into 20 unequal splits that simulate long-tailed distributions. **Right**: Our approach improves baseline performance even at higher difficulty.

guaranteed. For instance, baselines outperform DeCo-RN50 for the subsetting P→R in Office-Home, and SF(DA)$^2$+DeCo-RN101 shows a 9.9% decrease in accuracy for the `truck` class in VisDA. These fluctuations may indicate that ResNets make for poor references for debiasing. Their convolutional nature causes them to focus on spatially local features rather than global relationships, making them poor reference models for debiasing (Ye et al., 2024). However, HAcc results demonstrate that debiasing does occur and is beneficial to performance, even when using a ResNet smaller than the source backbone (see Appendix). This indicates that DeCo is able to mitigate source-specific bias but may not be able to address inductive bias (*i.e.*, learned during pre-training).

**Swin-B variants.** As expected, using a larger-scale backbone as the reference model leads to significant performance boosts. Swin-B has 3.5 times the number of parameters of ResNet-50 and twice of ResNet-101. This computational capacity and its hierarchical structure using shifted windows enables it to capture finer-grained nuances during pre-training (Liu et al., 2021b). DeCo-SwinB outperforms baselines in Office-Home (Table 1). As Hwang et al. (2024) does not report results for Office-Home, we provide reproduced results that are not optimized but still see +1.4% in average accuracy for SF(DA)$^2$. Particularly, we see notable improvements in harder tasks: A→C, P→C, and R→C. Respectively, our method outperforms the AaD baseline by 5.4%, 6.5%, and 7.2%. Results on DomainNet (Table 3) also show similar trends among clipart-related DA: DeCo-SwinB improves AaD by 6.1% in C→S, 5.6% in P→C, and 6.6% in R→C. DeCo also susbtantially boosts per-class accuracy for some (*i.e.*, `truck`) more so than others (*i.e.*, `horse`) in VisDA. We show that this increase is not solely due to the use of more advanced $f_r$ in HAcc evaluation.

**Debiasing.** We reiterate that HAcc measures debiasing in context to *source* bias, so gains in HAcc correspond to an increase in prediction accuracy from mitigating source-specific spurious cues. Figure 4 (right) illustrates the benefits of integrating DeCo-SwinB with the baseline AaD. Note that the difficulty of a balanced dataset can be interpreted as $1 - \frac{1}{C}$ and provides an intuitive reference point to assess split difficulty. For example, $h = 0.71$ makes split 1 elementary as $h < 0.92$. AaD+DeCo performs better than AaD as difficulty scales, with the largest gain (9%) seen in split 11, where $h = 0.99$. DeCo increases the average HAcc by 4.8%, an improvement that is more pronounced than the standard accuracy (*i.e.*, 2.4%). Our findings prove that debiasing aids performance even in the hardest splits (19 and 20), where the total number of samples is small and consist of samples from only 8 classes: `plane`, `bus`, `car`, `motorcycle`, `person`, `skateboard`, `train`, `truck`. However, gains are inconsistent for every split; the curve of AaD+DeCo begins to mimic that of AaD beyond split 11. This indicates that debiasing is effective up to a degree, perhaps only impacting certain classes that are absent in later splits. Also, we suspect that poor performance is also contributed by other factors such as visual ambiguity, severe occlusion, and object co-occurrence in the samples that DeCo cannot overcome. Even so, this observation suggests that debiasing – be it source-specific or not – is a *necessary but insufficient condition for better accuracy*. We include examples of ambiguous samples and supplementary results in the Appendix. Most significantly, we show that DeCo-RN50 boosts HAcc on split 11 by 4.4%.

**t-SNE.** We use t-Distributed Neighbor Embedding (t-SNE) (van der Maaten & Hinton, 2008) to visualize the feature spaces of split 11 for AaD and AaD+DeCo (Figure 5). While (a) already shows distinct clustering patterns, certain classes (*i.e.*, `motorcycle` and `car`) are still entangled.

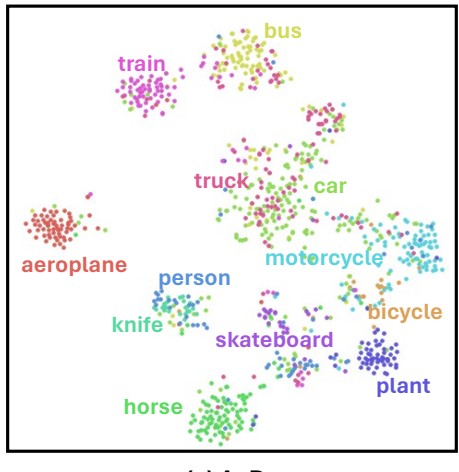 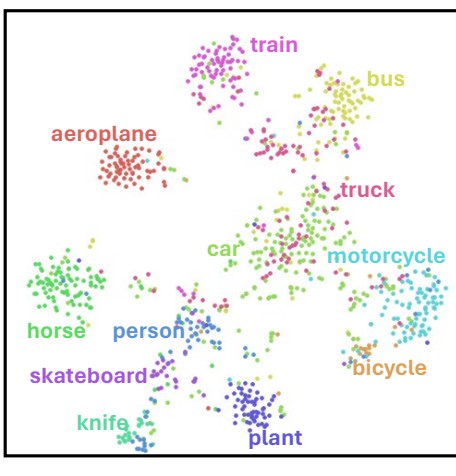

| (a) AaD | (b) AaD+DeCo |

Figure 5: t-SNE visualization of samples from split 11 ($h = 0.99$) as shown in Figure 4. Features are extracted at the last layer before the classification head. Samples are colored by class.

Our method succeeds in extracting more `car` samples from the `motorcycle` cluster. Another significant disentanglement is the separation of the `person` and `knife` clusters. We highlight that training with DeCo produces a feature space where feature similarity reflects semantic similarity. This can be observed among the `person`, `skateboard`, and `knife` clusters. As `skateboard` and `knife` co-occur frequently with `person`, it is intuitive that they would lie close to the `person` cluster in the feature space. In the same vein, although `car` and `truck` are still entangled in (b), there are discernibly less `truck` samples within the `car` cluster and more in `bus`. This aligns with the fact that trucks are more comparable in size to a bus than to a car.

## 5.3 ABLATION

**Similarity threshold.** Across all benchmarks, we keep the hyperparameters in DeCo identical except for $\beta_1$ and $\beta_2$, which act as the thresholds for samples considered similar and dissimilar, respectively. Ablations on both are in Figure 6 on Office-Home and VisDA. On all datasets, we follow the intuition that $\beta_1 \approx \frac{1}{C}$ and $\beta_2 \approx 1 - \frac{1}{C}$, where $C$ denotes the number of classes. In general, DeCo is not particularly sensitive to hyper-paramter changes.

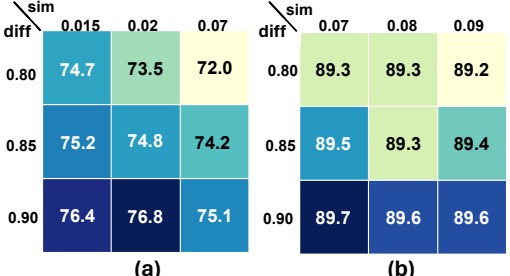

Figure 6: Ablations on $\beta_1$ and $\beta_2$ on Office-Home (a) and VisDA (b).

**Clustering objective.** Effective debiasing requires learning a strong cluster head that discriminates well. We replace $\mathcal{L}_g$ (Eq. 2) with other popular clustering methods and then train models with $\mathcal{L}_{aln}$. Table 6 reports results on VisDA (*i.e.*, average per-class accuracy) and AaD is used as the baseline. In addition to SimCLR (Chen et al., 2020) and Triplet loss (Hoffer & Ailon, 2018), we also test simpler approaches: (1) conducting mini-batch k-means clustering and computing the mean squared distance between features and the assigned cluster centroids (*i.e.*, K-means); and (2) using cluster assignments as pseudo-labels and calculating the cross-entropy loss (*i.e.*, CE). With a strong reference model, our method attains the best performance in terms of standard accuracy as well as HAcc.

**Number of MLP layers in $g$.** We ablate on the number of layers implemented in the multi-layer perceptron (MLP) projection head that precedes the final layer of the cluster head. Table 4 reports the average accuracy on VisDA when the number of MLP layers varies. We find that a lean MLP works best. In fact, removing the projector causes performance to drop – this is because post-backbone representations are only fully exploited

Table 4: Ablation on $g$.

| # MLP layers | Acc |
| --- | --- |
| 0 | 89.2 |
| 1 | 89.7 |
| 2 | 89.0 |
| 3 | 88.8 |

Table 6: Ablation on the clustering objective. Except for the baseline (*i.e.*, AaD), all objectives include training with $\mathcal{L}_{aln}$.

| Objective | Acc | HAcc (split 11) | HAcc (avg) |
|---|---|---|---|
| Baseline | 87.7 | 65.3 | 68.7 |
| SimCLR | 89.2 | 73.8 | 73.3 |
| Triplet | 87.8 | 62.9 | 64.9 |
| K-means | 87.8 | 65.5 | 69.5 |
| CE | 85.7 | 62.9 | 64.9 |
| $\mathcal{L}_g$ | 89.7 | 74.3 | 73.5 |

Table 7: Ablation on the technique used to determine feature similarity in DeCo.

| Dataset | Mode | Acc |
|---|---|---|
| Office-Home | stats | 75.7 |
| | sim | 76.1 |
| | prob | 76.8 |
| VisDA | stats | 87.4 |
| | sim | 89.1 |
| | prob | 89.7 |
| DomainNet | stats | 66.2 |
| | sim | 65.9 |
| | prob | 67.4 |

when high-quality pseudo-labels are available, and ost-projector features focus more on the pretext task (Wen et al., 2023).

**Contrastive modes in DeCo.** We experiment with different methods of determining feature similarity in Eq. 3. "prob" computes the dot product similarity with the softmaxed predictions of a given mini-batch, whereas "sim" uses the normalized predictions. The resulting similarity matrix is further refined based on $\beta_1$ and $\beta_2$. "stats" skips the refinement process and applies the matrix directly. Table 7 shows that "prob" mode consistently performs the best across all datasets. We believe that this is because softmax probabilities allow for direct probabilistic comparison between predictions, making it easier to interpret similarity.

Table 5: Ablation study on $\alpha$.

**Entropy weight coefficient.** The $\alpha$ weight coefficient in $\mathcal{L}_g$ represents the weight for the entropy regularizer, (*i.e.*, mean-entropy maximization). We follow SimGCD (Wen et al., 2023) and set $\alpha = 2.0$, but provide an ablation study in Table 5.

| Weight | Acc | HAcc |
|---|---|---|
| 0.5 | 89.5 | 73.5 |
| 1.0 | 89.4 | 73.1 |
| 2.0 | 89.7 | 74.3 |
| 5.0 | 89.6 | 72.5 |
| 10.0 | 89.6 | 72.5 |

### 5.4 Limitations & Technical Compute

We recognize the lack of and need for further study on the theoretical guarantee of debiasing SFDA. Additionally, our experiments only include studies on two baselines with DeCo. It would be beneficial to expand our investigation towards additional baselines, as well as debiasing for methods outside of neighborhood clustering. We focus our evaluations heavily on the VisDA dataset, as we believe that it is sufficiently challenging in comparison to Office-Home and DomainNet. More work on larger benchmarks of higher complexity is required to further validate our claims.

All experiments are run on a single NVIDIA RTXA6000. The runtime varies depending on the size of the dataset. For Office-Home, model training for each subtask usually takes less than two hours. It took about 10 hours to train on VisDA. DomainNet is much larger than Office-Home; the smallest subtask takes 5 hours and the largest can take up to 12 hours to train.

## 6 Conclusion

In this paper, we study the deficiency of current Source-Free Domain Adaptation (SFDA) methods in addressing source bias. We present a practical setting called Debiased SFDA as a solution, where model adaptation can leverage a pre-trained, frozen reference model. Building on this setting, we propose a simple approach, Debiased Contrastive learning (DeCo). The objective of DeCo is to align the source model with the clusters identified by the reference model, such that its classification boundary respects the cluster boundary. To evaluate its debiasing capabilities, we design a Hardness Accuracy (HAcc) metric that focuses on a model's performance on hard samples outside the source domain distribution. Extensive evaluations suggest that our proposed DeCo improves baselines on both accuracy and HAcc, especially with a strong reference model. As future work, we will study Debiased SFDA from a theoretical perspective and analyze its generalization bound.

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

# A APPENDIX

The appendix is organized as follows:

- Section A.1 reports results on experiments regarding the reference model. It also highlights the importance of HAcc in assessing model performance under strong bias.
- Section A.2 provides additional information on our proposed unsupervised clustering objective, $\mathcal{L}_g$, such as the data augmentations used, ablation studies, and observations on uniformity.
- Section A.3 describes the diagnostic experiment mentioned in the Introduction.
- Section A.4 details how evaluative $K$-partitions are created to measure a model's HAcc.
- Section A.5 presents examples of visual ambiguity in the VisDA dataset.

Table 8: List of abbreviations and symbols used in the paper.

| Abbreviation/Symbol | Meaning |
|---|---|
| *Abbreviation* | |
| DA | Domain Adaptation |
| UDA | Unsupervised Domain Adaptation |
| SFDA | Source-Free Domain Adaptation |
| HAcc | Hardness Accuracy |
| *Symbol in Algorithm* | |
| $S$ | Source domain |
| $T$ | Target domain |
| $\mathbf{x}$ | Sample feature |
| $\mathcal{K}$ | Set of $k$ nearest neighbors |
| $N$ | Number of samples in $T$ |
| $f_s$ | Source pre-trained model |
| $f_r$ | Frozen reference model |
| $g$ | Cluster head |
| $\mathcal{L}_{adp}$ | Adaptation loss |
| $\mathcal{L}_{reg}$ | Regularization loss |
| $\mathcal{L}_{aln}$ | Alignment loss |

## A.1 CHOICE OF REFERENCE MODEL

Larger and more complex models are likely to outperform smaller counterparts in classification tasks as they have an increased ability to capture visual nuances. Thus, using a large pre-trained foundation model as $f_r$ is quintessential in achieving strong results. However, we demonstrate that using smaller models with DeCo still conducts debiasing to provide performance gains, especially on biased datasets. Table 9 reports the results of our experiments.

HAcc shows the benefits of DeCo more clearly – DeCo-RN50 increases HAcc in split 11 by 4.4% and the overall average by 1.5%. Similarly, DeCo-SwinT leads to a 3.1% improvement in average HAcc. This indicates that smaller, less complex models still provide useful clues for mitigating source bias. Additionally, we observed that Swin variants of DeCo outperform those of ResNet. Swin-Tiny (28.3m parameters) is comparative to ResNet-50 (25.6m) in size and smaller than the ResNet-101 (44.5m), which is the standard visual backbone used in VisDA evaluation. Notably, DeCo-SwinT achieves stronger performance than DeCo-RN101 on the standard benchmark. This improvement can be attributed to the differences in architecture — while ResNet models are efficient and effective at capturing local features and spatial hierarchies, Swin transformers excel at modeling global context and long-range dependencies. The representation power of Swin transformers may indicate less likelihood to rely on spurious correlations to make predictions, thereby acting as better references for the adapting model.

Table 9: We implement reference models of ranging parameter size and report the average per-class accuracy on VisDA, as well as the HAcc (split 11 and average).

| Model | Acc | HAcc (split 11) | HAcc (avg) |
|---|---|---|---|
| Baseline | 87.7 | 65.3 | 68.7 |
| +DeCo-RN50 | 88.3 | 69.7 | 70.2 |
| +DeCo-RN101 | 88.4 | 70.1 | 70.5 |
| +DeCo-SwinT | 89.4 | 72.9 | 71.8 |
| +DeCo-SwinS | 89.4 | 73.1 | 72.6 |
| +DeCo-SwinB | 89.7 | 74.3 | 73.5 |

### A.2 UNSUPERVISED CLUSTERING OBJECTIVE $\mathcal{L}_g$

#### A.2.1 DATA AUGMENTATIONS

For data augmentations, training samples are resized to $256 \times 256$, randomly cropped to $224$, then random horizontal flip is applied. We implement a strong augmented view of $\mathbf{x}$ in addition to the standard augmentation, which we consider the weakly augmented view. Strong augmentations include the same transformations as the original, but further includes a random application of color jitter, as well as RandAugment (Cubuk et al., 2019).

#### A.2.2 UNIFORM DISTRIBUTION IN CLUSTERS

$\mathcal{L}_g$ avoids the possibility of uniform distribution (*i.e.*, $\frac{1}{C}, \frac{1}{C}, ..., \frac{1}{C}$) by using sharpened target predictions and mean-entropy maximization, $H(\bar{p})$. We utilize an exponential moving average-updated teacher model to sharpen soft cluster pseudo-labels for loss calculation. Sharper target predictions guide the model to produce confident low entropy anchor predictions; they also make them unequal to the uniform distribution. Additionally, $H(\bar{p})$ seeks to minimize the negative entropy of the average prediction across all anchor views in a given mini-batch. In using sharpened target predictions, the average prediction used in entropy regularization is also not equal to the uniform distribution. Further information on the proof is provided in the appendix of Assran et al. (2022).

### A.3 DIAGNOSTIC EXPERIMENT

We create easy and hard splits using the VisDA dataset, where $S_{easy}$ contains samples with features that are most similar to the source domain, and $S_{hard}$ contains those most dissimilar. Note that the sets are identical in size and do not overlap. Similarity is determined by the cosine similarity between the mean class representation of $S$ and a given sample in $T$, as evaluated by the source pre-trained model $f_s$. This means that sample similarity is measured in relation to source bias. Then, we run inference with models trained on the full dataset on these splits. We experiment with varying the size of the split by taking the top-$K$ target test samples of highest/lowest similarity to the source domain *for every class*; the resulting split is balanced in class distribution. We report the average per-class accuracy for two baselines in Table 10.

### A.4 EVALUATIVE PARTITIONS

To assess a model's robustness towards source bias and simulate the difficulty of real-world scenarios, we aim to generate datasets that simulate long-tailed distribution for testing. Specifically, we experiment with using the Power Law (Clauset et al., 2009) and Zipf's Law (Powers, 1998) to calculate the hardness ratio, which is used to determine the composition of difficult and easy samples in a given split. For the Power Law, we set $\alpha_{pow} = 6.0$ based on the size of the dataset and generate 20 partitions of unequal length and class distribution. We also tried partitioning using Zipf's law by ranking all $T$ samples based on the defined hardness score to create 200 partitions. We proceed default to using the Power Law as it illustrates performance dichotomy more prominently than Zipf's law.

Table 10: Per-class accuracy on VisDA easy/hard splits (%). We implement DeCo with Swin-B as the reference model.

| Split | AAD | AAD + DECO | SF(DA)$^2$ | SF(DA)$^2$ + DECO |
|-------|-----|------------|------------|-------------------|
| Easy-50 | 97.5 | 98.5 | 94.7 | 98.3 |
| Easy-300 | 95.3 | 96.6 | 92.5 | 96.3 |
| Easy-500 | 94.5 | 95.8 | 91.6 | 95.6 |
| Hard-50 | 61.2 | 68.3 | 63.5 | 67.7 |
| Hard-150 | 66.3 | 70.9 | 68.6 | 70.7 |
| Hard-200 | 68.6 | 73.1 | 70.5 | 72.3 |
| Hard-300 | 70.8 | 75.4 | 73.0 | 74.6 |

## A.5 AMBIGUOUS SAMPLES

We provide qualitative examples of images from the VisDA dataset that are difficult to assess due to visual ambiguity, severe occlusion, or object co-occurrence. The samples are center-cropped instead of randomly cropped to maximize salient object information, as subjects are generally at the center of the image.

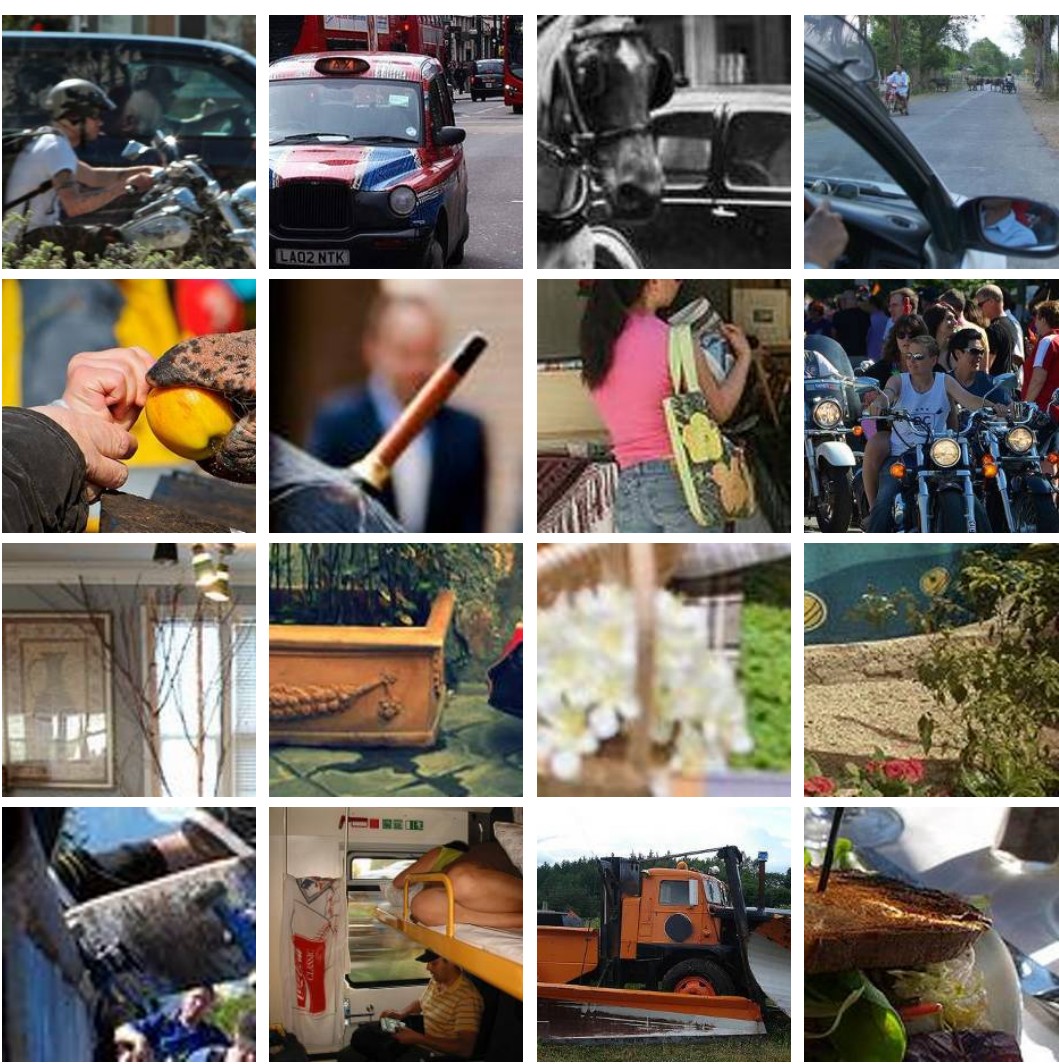

Figure 7: Samples from VisDA that are visually ambiguous. The ground-truth labels for the first, second, and third rows are: car, person, and plant. We include additional samples in the final row with the following ground-truth labels: skateboard, train, truck, and knife.

