# OpenReview forum: "Towards Debiased Source-Free Domain Adaptation"
_ICLR.cc/2025/Conference — ICLR 2025 Conference Withdrawn Submission_

### Official Review · Reviewer_L7Xc · 2024-10-25

**Soundness:** 2
**Presentation:** 3
**Contribution:** 3
**Rating:** 5
**Confidence:** 3

**Summary:**

This manuscript studies the source-free domain adaptation (SFDA) problem and aims to address the issue of source bias in the adaptation process. The authors point out that spurious correlations and the absence of methods to handle source bias under SFDA make it difficult for SFDA algorithms to ensure good model performance in the target domain. To address this, the manuscript proposes Debiased SFDA by leveraging a reference model trained on a large and diverse dataset as a debiasing guide. The goal is to discard source-specific traits and retain more generalizable knowledge during the adaptation process. Based on this motivation, the authors introduce a contrastive learning objective (DeCo) to debias the source-trained model and propose a diagnostic metric (Hardness Accuracy) to assess the degree of source bias in the adapted model. Empirical evaluations are conducted on standard benchmarks to demonstrate the performance improvements from the proposed method when integrated with existing SFDA frameworks.

**Strengths:**

1. This manuscript raised concerns about the source bias problem in the SFDA problem, which is crucial and practical in real-world scenarios.

2. The proposed method is simple to implement and can be integrated with existing SFDA methods.

3. The authors tried to introduce a quantitative and diagnostic metric, Hardness Accuracy (HAcc), to measure the degree of debiasing, which can be valuable.

**Weaknesses:**

1. This manuscript proposed to debias the source model with a reference model that is pre-trained on large and diverse datasets. However, I still have concerns regarding this extra model because it brings another problem about how to guarantee the debiasing extent of this reference model. Unfortunately, the quality of the reference model has not been discussed in the current manuscript.

2. Only two SFDA methods were integrated with the proposed debiasing algorithm proposed in this manuscript. The results were a little weak to support its effectiveness.

**Questions:**

1. The transition between the second and third paragraphs of the Introduction part seems confusing. In Line046, the authors claimed that "The use of such spurious correlations may cause samples of different classes to have the same prediction, as they are erroneously considered among the nearest neighbors".  Then, in the third paragraph of the introduction, the authors stated some experimental observations w.r.t. the **confident correct samples** (i.e., easy split) and **confident wrong samples** (i.e., hard split). However, I did not see a clear connection between the motivation and the observations of such experiments. I understand that Appendix A.3 provided an explanation, but I believe that the introduction part needs to be further improved, especially needs to give straightforward support to your claim at Line046.

2. About the relationship between the class spaces of SFDA (i.e., within S/T domains) and reference pre-training datasets (e.g., ImageNet). Between Line103 and Line104, the authors claimed that "(b) classes in T can be considered novel to $f_r$, as it is pre-trained on a different dataset". However,  in Lines 210-212, the authors also stated that "2) Its pre-training dataset should be diverse to learn general knowledge, while also related to the SFDA task to allow knowledge transfer". The above two expressions seem contradictory. I wonder what kind of datasets that we should expect to pre-train the reference model $f_r$. In this manuscript, the reference model is pre-trained on ImageNet dataset, whose label space has some overlaps with that of Office-Home, VisDA and DomainNet datasets. How about using a pre-training dataset that has no overlap with these test sets?  Could the author provide more ablation studies w.r.t. different pre-training datasets? I believe this is also crucial to understand the proposed method.

3. Based on the above point, we can consider a more general question: how to guarantee the debiasing extent of the reference model such that it is debiased enough to be leveraged by the source model? I believe that the reference model may also be biased if we cannot choose the right one. However, this point has not been discussed in the current manuscript.

4. I noticed that only two methods (i.e., AaD and SF(DA)^2) were integrated with the proposed debiasing algorithm. Could the authors introduce and discuss more methods to support the effectiveness of their algorithm?

5. Some notations used in this manuscript are confusing:
	- $L_{aad}$ in Eq. (1) and $L_{AaD}$ in Eq.(4) are not consistent;
	- What is $f_g$ in Line268? Do you mean $f_g=g \circ f_r$?
	- Between Line277 and Line282, I wonder if $g(\mathbf{x})$ here indicates $g(f_r(\mathbf{x}))$?

---

### Official Review · Reviewer_2HxY · 2024-10-31

**Soundness:** 2
**Presentation:** 3
**Contribution:** 2
**Rating:** 5
**Confidence:** 3

**Summary:**

The paper deals with an important task of source-free domain adaptation. The authors argue that due to the learning of spurious features in the source, the model is often unable to transfer well onto a new target domain with a significant domain gap. This is also described as the bias of the learned model by the authors. They demonstrate the bias by a diagnostic experiment that shows the model is unable to generalize on the hard split of the test data as it generalises on the easy split with more adaptation steps, indicating a precision-recall kind of trade-off. DeCo is proposed as a solution to de-bias the model with the help of a larger pre-trained model. The method employs contrastive loss training on the test domain data to refine the source model and de-bias it. The designed debiasing objective is then benchmarked on several datasets. The authors have also proposed a hardness accuracy to measure an adapted model’s accuracy in relation to initial source bias.

**Strengths:**

+ Tackles an important instance of domain adaptation (source-free domain adaptation)
+ Problem setting in the paper is well-motivated and easy to follow
+ Authors position the differences of their work well in contrast to previous works in source-free domain adaptation
+ Extensive experiments and rigorous evaluation of the proposed method DeCo

**Weaknesses:**

- While the proposed method is a good application of existing solutions (contrastive loss and additional teacher model), unlike the problem setting, the proposed method is not well motivated and lacks novelty.
- Motivation for the definition of Hardness accuracy is not clear (see questions). This motivation becomes very important since authors argue that although the model does only incrementally perform better on the benchmark, it has been debiased (in context to source bias) as indicated by lower Hardness scores.
- Lack of theoretical motivation. There is no discussion on the generalisation performance of SFDA. It would be interesting for authors to connect the DeCo to the invariance principle (IRM) [1,2] and other causal domain generalisation literature. This can justify DeCo as a transferer of stable non-spurious features from source to target.

[1] Arjovsky, Martin, et al. "Invariant risk minimization." arXiv preprint arXiv:1907.02893 (2019).
[2] Ahuja, Kartik, et al. "Invariant risk minimization games." International Conference on Machine Learning. PMLR, 2020.

**Questions:**

+ Regarding H-score: While the H-score computes the probability mass on incorrect classes. As the predictions become more underconfident, the H-score can still decrease. However, the entropy on the test domain with further adaptation epochs will increase or plateau if accuracy is not improving with DeCo as indicated in the benchmarks. Authors should include entropy curves also in their analysis.
+ It would be interesting to have a detailed ablation of how much the accuracy of the large-scale pre-trained model on the test domain affects DeCo. In a hypothetical setup, is DeCo able to match the performance of the open-source model?
+ Does DeCo debias the source model to leave only invariant features? It would interesting to compare DeCo to an invariant risk minimizer across domains.

---

### Official Review · Reviewer_VR5i · 2024-11-04

**Soundness:** 2
**Presentation:** 2
**Contribution:** 2
**Rating:** 3
**Confidence:** 5

**Summary:**

This paper introduces a pre-trained and frozen reference model into the original Source-Free Domain Adaptation (SFDA) setting, called Debiased SFDA. In this context, the reference model provides source-domain-independent debiased prior knowledge. The authors then use the reference model as a pre-trained feature mapper for unsupervised clustering training on target domain samples and introduce a contrastive loss to align the source model's classification boundaries and clustering boundaries. The experiments validate the effectiveness of the proposed method.

**Strengths:**

1.	The authors discover a consistent trade-off in the performance of existing SFDA methods on easy samples (with small inter-domain feature differences) versus difficult samples (with large inter-domain feature differences) through a preliminary experiment, which is a novel finding.
2.	The authors split the dataset based on a hardness score following a power-law distribution, which can evaluate the effectiveness of the proposed method under different scenarios with varying source-target domain differences.
3.	The proposed method demonstrates a certain level of effectiveness, with experiments showing that it generally outperforms existing SFDA models. It can also be integrated as a plug-and-play component into other methods for debiasing purposes.

**Weaknesses:**

1.	The novelty of this work is relatively limited. This study adds an extra pre-trained model to SFDA and fine-tunes the source domain-trained model using a basic contrastive loss. This introduces prior knowledge beyond the source and target domains, which the authors attribute this advantage to debiasing. The improvement in the adapted model's performance seems predictable.
2.	The technical contribution of this work appears insufficient. The proposed method consists of two steps: (1) removing the classification head of the reference model and unsupervised training of a new clustering head; (2) aligning the source domain model and clustering boundaries through contrastive loss. Step one uses existing mature methods, and step two uses a contrastive loss, where positive and negative pairs are selected based on the similarity of clustering probability (inner products.)

**Questions:**

Questions
1.	On page 5, line 225, "model" is misspelled as "mdoel".
2.	I can understand that freezing the reference model and training the clustering head from scratch, which is independent of the source domain model, can prevent the clustering model from acquiring incorrect knowledge from the source domain model. However, when the reference model is poor, clustering will produce many indistinguishable samples, and simply matching the classification boundaries of the source domain model may introduce errors. In particular, the selection of positive and negative pairs in the contrastive loss completely relies on the clustering model, which may further exacerbate the mistakes.
3.	In the standard InfoNCE method, the denominator of the contrastive loss includes positive sample pairs. Note that the denominator in Eq. (3) only includes negative sample pairs. What is the advantage of this design?
4.	On page 5, line 268, the symbol $f_g$ appears without explanation.
5.	The authors split the dataset based on hardness scores, generating a series of sample sets with varying degrees of domain bias to assess the domain adaptation performance under different biases. However, the authors' claim on page 3, line 114, that they propose "a novel HAcc metric to assess source bias" is inaccurate. HAcc is used to evaluate the average effect of domain adaptation methods under different domain bias scenarios, rather than "assess source bias". This representation is worth adjusting.
6.	As one of the contributions highlighted by the authors in the introduction, the manuscript's description of H-score and HAcc is insufficiently detailed. The authors mention on page 7, line 367, "Additional details are provided in the Appendix", but the description in the appendix is still unclear. Moreover, note that most experiments on HAcc are conducted on the VisDA dataset, and more experiments on other datasets may help evaluate the generality of this metric.
7.	Figure 6 does not explicitly label $\beta_1$ and $\beta_2$. I suggest modifying the figure to make it more intuitive.

---

### Official Review · Reviewer_BFy4 · 2024-11-05

**Soundness:** 2
**Presentation:** 2
**Contribution:** 2
**Rating:** 5
**Confidence:** 4

**Summary:**

In this work, the authors study source-free domain adaptation (SFDA) from the perspective of spurious correlation and source bias. Specifically, they propose a Debiased Contrastive Learning (DeCo) method, utilizing additional supervision from a pre-trained and frozen reference model to reduce source bias in SFDA while enhancing adaptation performance. Additionally, the authors introduce a novel metric, HAcc, to quantify the debiasing effect. They conduct extensive experiments and analyses to validate their proposed method.

**Strengths:**

- This paper investigates SFDA with a focus on spurious correlations and bias, presenting an interesting perspective.

- The proposed method shows effectiveness based on experimental results, with reasonable experiments and corresponding analysis.

- The paper is well-organized and structured.

**Weaknesses:**

- **Debiasing Concept**: The introduction of the debiasing concept lacks clarity. While the introduction addresses spurious correlations learned by the source model in the target domain, it lacks a clear and comprehensive definition of source bias. Spurious correlation in SFDA has also been studied previously, though within different neural network frameworks [1].

- **Efficiency Analysis**: There is a lack of analysis and comparison regarding efficiency. Since DeCo introduces external features and an additional network structure (g), as well as data augmentations, computational consumption, memory usage, and training time should be further compared or discussed.

- **External Feature Extractors**: While using external feature extractors improves performance, it is not a surprising approach, as it incorporates external training knowledge and has been studied in recent work [2, 3].


[1] Sunandini Sanyal, Ashish Ramayee Asokan, Suvaansh Bhambri, Pradyumna YM, Akshay Kulkarni, Jogendra Nath Kundu, R. Venkatesh Babu: Aligning Non-Causal Factors for Transformer-Based Source-Free Domain Adaptation. 2024 CVPR.

[2] Idit Diamant, Idan Achituve, Arnon Netzer: De-Confusing Pseudo-Labels in Source-Free Domain Adaptation. 2024 ICCV.

[3] Wenyu Zhang, Li Shen, Chuan-Sheng Foo: Rethinking the Role of Pre-Trained Networks in Source-Free Domain Adaptation. 2023 ICCV.

**Questions:**

- **Methodology and Experiment**: In Eq. (3), can the alignment term be combined with the AaD method, potentially replacing the infoNCE loss with the dot product in AaD? Additionally, what are the final distributions of positive and negative neighbor counts across different datasets?

- **Expression**:

    - The definitions of H-scores and HAcc, along with Figure 4 and the debiasing analysis, are somewhat confusing. For example, in lines 323 and 364, it is stated that the subsequent split contains fewer samples with higher H-scores, where higher H-scores represent samples that are more distinct or challenging compared to the source domain. Shouldn't the earlier splits be more difficult? Furthermore, in the right panel of Figure 4, what does "h" represent? Does it refer to HAcc (defined as mean-class accuracy in line 365) or the average H-score for a specific split? These details make the debiasing section somewhat unclear.

    - **Motivation**: Figure 1 illustrates the varying performances of different SFDA methods on easy and hard samples, defined by the similarity between target data points and the source model’s average representation. This performance gap suggests the presence of distribution shift, but how does it relate to spurious correlation or source bias? Please correct me if I’ve misunderstood.

---

### Note · Authors · 2024-11-14

**Comment:**

We appreciate the constructive feedback that reviewers have taken the time to provide us and will make appropriate revisions accordingly. Thank you!

**Withdrawal Confirmation:**

I have read and agree with the venue's withdrawal policy on behalf of myself and my co-authors.